# RELATIONAL OBJECT-CENTRIC ACTOR-CRITIC

## ABSTRACT

There have recently been significant advances in the problem of unsupervised object-centric representation learning and its application to downstream tasks. The latest works support the argument that employing disentangled object representations in image-based object-centric reinforcement learning tasks facilitates policy learning. We propose a novel object-centric reinforcement learning algorithm combining actor-critic and model-based approaches, by incorporating an object-centric world model in critic. The proposed method fills a research gap in developing efficient object-centric world models for reinforcement learning settings that can be used for environments with discrete or continuous action spaces. We evaluated our algorithm in simulated 3D robotic environment and a 2D environment with compositional structure. As baselines, we consider the state-of-the-art model-free actor-critic algorithm built upon transformer architecture and the state-of-the-art monolithic model-based algorithm. While the proposed method demonstrates comparable performance to the baselines in easier tasks, it outperforms the baselines within the 1M environment step budget in more challenging tasks increased number of objects or more complex dynamics.

## 1 INTRODUCTION

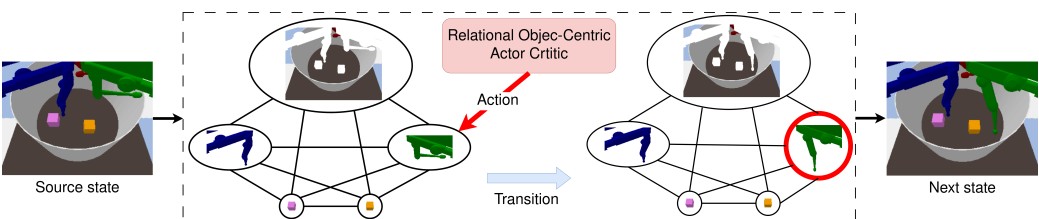

Figure 1: A high-level overview of the proposed method. ROCA learns the policy by extracting object-centric representations from the source image and treating them as a complete graph.

One of the primary problems in visual-based reinforcement learning is determining how to represent the environment's state efficiently. The most common approach is to encode the entire input image, which is then used as input for the policy network (Mnih et al., 2015; Zhang et al., 2021). However, previous studies (Santoro et al., 2017) have shown that such representations may fail to capture meaningful relationships and interactions between objects in the state. Object-centric representations can be introduced to overcome this issue. Such representations are expected to result in more compact models with enhanced generalization capabilities (Keramati et al., 2018). State-of-the-art unsupervised object-centric representation (OCR) models (Singh et al., 2022; Locatello et al., 2020b; Engelcke et al., 2022) have a fundamental appeal for RL as they do not require additional data labeling for training. Recent studies (Stanić et al., 2022; Yoon et al., 2023) have shown that object-centric state factorization can improve model-free algorithms' generalization ability and sample efficiency.

Another way to reduce the number of necessary environment samples is to use model-based methods (Sutton & Barto, 2018). In model-based reinforcement learning (MBRL), the agent constructs models for transition and reward functions based on its experience of interaction with the environment. The agent performs multi-step planning to select the optimal action using the model's

predictions without interacting with the environment. The model-based algorithms could be more efficient than model-free algorithms if the accuracy of the world model is sufficient. State-of-the-art MBRL methods, employing learning in imagination (Hafner et al., 2023) and lookahead search with value equivalent dynamics model (Ye et al., 2021) master a diverse range of environments.

To further enhance sample efficiency, a promising direction is to combine both approaches by developing a world model that leverages object representations and explicitly learns to model relationships between objects (Zholus et al., 2022). An example of this approach is the contrastively-trained transition model CSWM (Kipf et al., 2020). It uses a graph neural network to approximate the dynamics of the environment and simultaneously learns to factorize the state and predict changes in the state of individual objects. CSWM has shown superior prediction quality compared to traditional monolithic models.

However, OCR models demonstrate high quality in relatively simple environments with strongly distinguishable objects (Wu et al., 2023). Additionally, in object-structured environments, actions are often applied to a single object or a small number of objects, simplifying the prediction of individual object dynamics. In more complex environments, the world model must accurately bind actions to objects to predict transitions effectively. Despite recent progress (Biza et al., 2022), no fully-featured dynamics models considering the sparsity of action-object relationships have been proposed. These challenges make it difficult to employ object-centric world models in RL. For instance, the CSWM model has not been utilized for policy learning in offline or online settings.

Our research is focused on value-based MBRL as object-based decomposition of value function could contribute to the training of object-centric world model consistent with policy. We introduce the Relational Object-Centric Actor-Critic (ROCA), an off-policy object-centric model-based algorithm inspired by the Soft Actor-Critic (SAC) (Haarnoja et al., 2018; 2019; Christodoulou, 2019) that operates with both discrete and continuous action spaces. The ROCA algorithm uses the pre-trained SLATE model (Singh et al., 2022), which extracts representations of the individual objects from the input image. Similar to CSWM (Kipf et al., 2020), we utilize a structured transition model based on graph neural networks. Our reward, state-value, and actor models are graph neural networks designed to align with the object-centric structure of the task. Inspired by TreeQN (Farquhar et al., 2018), we use a world model in the critic module to predict action-values. The ROCA algorithm is the first to apply a GNN-based object-centric world model for policy learning in the RL setting successfully. To evaluate the algorithm's quality, we conducted experiments in 2D environments with simple-shaped objects and visually more complex simulated 3D robotic environments. The proposed algorithm demonstrates high sample efficiency and outperforms the object-oriented variant of the model-free PPO algorithm (Schulman et al., 2017), which uses the same SLATE model as a feature extractor and is built upon the transformer architecture. Furthermore, our method performs better than the state-of-the-art MBRL algorithm DreamerV3 (Hafner et al., 2023).

Our contributions can be summarized as follows:

- We propose a novel architecture that combines a value-based model-based approach with the actor-critic SAC algorithm by incorporating a world model into the critic module.
- We extended the SAC algorithm by introducing a new objective function to train the model-based critic.
- We propose a GNN-based actor to pool object-centric representations.
- We modified the GNN-based CSWM transition model by adjusting its edge model: we pass a pair of slots along with an action into the edge model.

## 2 RELATED WORK

**Object-Centric Representation Learning** Recent advancements in machine learning research have been dedicated to developing unsupervised OCR algorithms (Ramesh et al., 2021; Locatello et al., 2020b; Engelcke et al., 2022). These methods aim to learn structured visual representations from images without relying on labeled data, modeling each image as a composition of objects. This line of research is motivated by its potential benefits for various downstream tasks, including enhanced generalization and the ability to reason over visual objects. One notable approach in

this field is Slot-Attention (Locatello et al., 2020b), which represents objects using multiple latent variables and refines them through an attention mechanism. Building upon this, SLATE (Ramesh et al., 2021) further improves the performance by employing a Transformer-based decoder instead of a pixel-mixture decoder.

**Object-Centric Representations and Model-Free RL**    Stanić et al. (2022) uses Slot-Attention as an object-centric feature extractor and examines the performance and generalization capabilities of RL agents. In another study (Sharma et al., 2023), a multistage training approach is proposed, involving fine-tuning a YOLO model (Jocher et al., 2022) on a dataset labeled by an unsupervised object-centric model. The frozen YOLO model is then employed as an object-centric features extractor in the Dueling DQN algorithm. Object representations are pooled using a graph attention neural network before being fed to the Q-network.

**Object-Centric Representations and MBRL**    As related work in object-oriented MBRL, we consider Watters et al. (2019). It uses MONet (Burgess et al., 2019) as an object-centric features extractor and learns an object-oriented transition model. However, unlike our approach, this model does not consider the interaction between objects and is only utilized during the exploration phase of the RL algorithm.

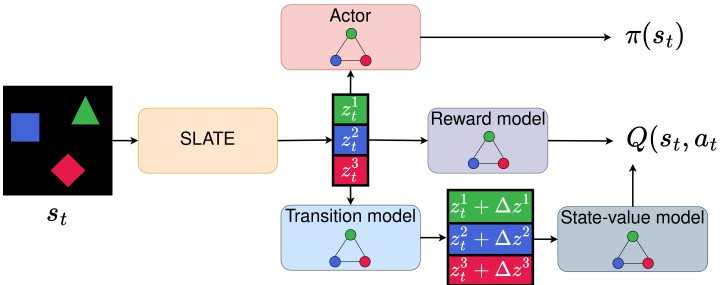

Figure 2: ROCA overview. Framework consists of a pre-trained frozen SLATE model, which extracts object-centric representations from an image-based observation, and GNN-based modules: a transition model, a reward model, a state-value model, and an actor model. The transition and reward models form a world model. The world model and the state-value model together constitute the critic module, which predicts Q-values.

# 3    BACKGROUND

## 3.1    MARKOV DECISION PROCESS

We consider a simplified version of the object-oriented MDP (Diuk et al., 2008):

$$\mathcal{U} = (\mathcal{S}, \mathcal{A}, T, R, \gamma, \mathcal{O}, \Omega), \tag{1}$$

where $\mathcal{S} = \mathcal{S}_1 \times \cdots \times \mathcal{S}_K$ — a state space, $\mathcal{S}_i$ — an individual state space of the object $i$, $\mathcal{A}$ — an action space, $T = (T_1, \ldots, T_K)$ — a transition function, $T_i = T_i(T_{i1}(s_i, s_1, a), \ldots, T_{iK}(s_i, s_K, a))$ — an individual transition function of the object $i$, $R = \sum_{i=1}^{K} R_i$ — a reward function, $R_i = R_i(R_{i1}(s_i, s_1, a), \ldots, R_{iK}(s_i, s_K, a))$ — an individual reward function of the object $i$, $\gamma \in [0; 1]$ — a discount factor, $\mathcal{O}$ — an observation space, $\Omega : \mathcal{S} \rightarrow \mathcal{O}$ — an observation function. The goal of reinforcement learning is to find the optimal policy: $\pi^* = \arg\max_\pi \mathbb{E}_{s_{t+1} \sim T(\cdot|s_t, a_t), a_{t+1} \sim \pi(\cdot|s_{t+1})} \left[ \sum_{i=0}^{\tau} \gamma^t R(s_t, a_t) \right]$ for all $s_0$ where $\tau$ is the number of time steps.

In model-based approach the agent uses the experience of interactions with the environment to build a world model that approximates the transition function $\hat{T} \approx T$ and the reward function $\hat{R} \approx R$ and use its predictions as an additional signal for policy learning.

## 3.2 Soft Actor-Critic

Soft Actor-Critic (SAC) (Haarnoja et al., 2018; 2019) is a state-of-the-art off-policy reinforcement learning algorithm for continuous action settings. The goal of the algorithm is to find a policy that maximizes the maximum entropy objective:

$$\pi^* = \text{argmax}_\pi \sum_{i=0}^{\tau} \mathbb{E}_{(s_t, a_t) \sim d_\pi} \left[ \gamma^t (R(s_t, a_t) + \alpha \mathcal{H}(\pi(\cdot|s_t))) \right]$$

where $\alpha$ is the temperature parameter, $\mathcal{H}(\pi(\cdot|s_t)) = -\log \pi(\cdot|s_t)$ is the entropy of the policy $\pi$ at state $s_t$, $d_\pi$ is the distribution of trajectories induced by policy $\pi$. The soft action-value function $Q_\theta(s_t, a_t)$ parameterized using a neural network with parameters $\theta$ is trained by minimizing the soft Bellman residual:

$$J_Q(\theta) = \mathbb{E}_{(s_t, a_t) \sim D} \left[ \left( Q_\theta(s_t, a_t) - R(s_t, a_t) - \gamma \mathbb{E}_{s_{t+1} \sim T(s_t, a_t)} V_{\bar{\theta}}(s_{t+1}) \right)^2 \right] \quad (2)$$

where $D$ is a replay buffer of past experience and $V_{\bar{\theta}}(s_{t+1})$ is estimated using a target network for $Q$ and a Monte Carlo estimate of the soft state-value function after sampling experiences from the $D$.

The policy $\pi$ is parameterized using a neural network with parameters $\phi$. The parameters are learned by minimizing the expected KL-divergence between the policy and the exponential of the $Q$-function:

$$J_\pi(\phi) = \mathbb{E}_{s_t \sim D} \left[ \mathbb{E}_{a_t \sim \pi_\phi(\cdot|s_t)} \left[ \alpha \log(\pi_\phi(a_t|s_t)) - Q_\theta(s_t, a_t) \right] \right] \quad (3)$$

The objective for the temperature parameter is given by:

$$J(\alpha) = \mathbb{E}_{a_t \sim \pi(\cdot|s_t)} \left[ -\alpha(\log \pi(a_t|s_t) + \bar{H}) \right] \quad (4)$$

where $\bar{H}$ is a hyperparameter representing the target entropy. In practice, two separately trained soft Q-networks are maintained, and then the minimum of their two outputs are used to be the soft Q-network output.

While the original version of SAC solves problems with continuous action space, the version for discrete action spaces was suggested by Christodoulou (2019). In the case of discrete action space, $\pi_\phi(a_t|s_t)$ outputs a probability for all actions instead of a density. Such parametrization of the policy slightly changes the objectives 2, 3 and 4. We describe SAC in more details in appendix B.

## 4 Relational Object-Centric Actor-Critic

Figure 2 outlines the high-level overview of the proposed actor-critic framework (ROCA). As an encoder we use SLATE (Singh et al., 2022), a recent object-centric model. SLATE incorporates a dVAE (van den Oord et al., 2018) for internal feature extraction, a GPT-like transformer (Ramesh et al., 2021) for decoding, and a slot-attention module (Locatello et al., 2020c) to group features associated with the same object. We refer to appendix C for a more detailed description of SLATE. In ROCA the pre-trained frozen SLATE model takes an image-based observation $s_t$ as input and produces a set of object vectors, referred to as slots, $\boldsymbol{z}_t = (z_t^1, \ldots, z_t^K)$ ($K$ - the maximum number of objects to be extracted). An actor model encapsulates the current agent's policy and returns an action for the input state $\boldsymbol{z}_t$. Critic predicts the value $Q(\boldsymbol{z}_t, a)$ of the provided action $a$ sampled from the actor given the current state representations $\boldsymbol{z}_t$. It is estimated using the learned transition model, reward model, and state-value model. The input state representation $\boldsymbol{z}_t = (z_t^1, \ldots, z_t^K)$ is treated as a complete graph while being processed by GNN-based components of the ROCA.

### 4.1 Transition Model

We approximate the transition function using a graph neural network Kipf et al. (2020) with an edge model $\text{edge}_T$ and a node model $\text{node}_T$ which takes a factored state $\boldsymbol{z}_t = (z_t^1, \ldots, z_t^K)$ and action $\boldsymbol{a}_t$ as input and predicts changes in factored states $\Delta \boldsymbol{z}$. The action is provided to the node model $\text{node}_T$ and the edge model $\text{edge}_T$ as shown in Figure 3. The factored representation of the next

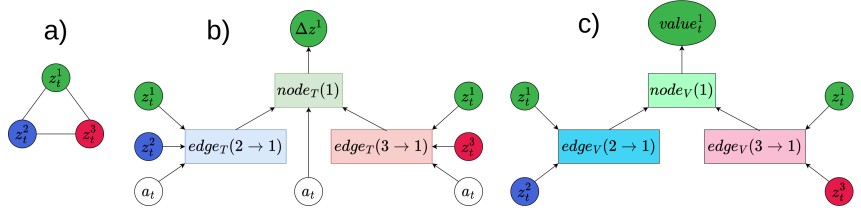

Figure 3: Overview of GNN-based transition model and state-value model. **a)** Representation of the state as a complete graph. **b)** Transition model: message-passing update scheme for the embedding of object 1. **c)** State-value model: message-passing update scheme for the state-value prediction for the object 1.

state is obtained via $\hat{z}_{t+1} = z_t + \Delta z$. Since we treat the set of slots as a complete graph, the complexity of the update rule 5 is quadratic in the number of slots. The same applies to all GNN models in the ROCA.

$$\Delta z^i = \texttt{node}_T(z_t^i, a_t^i, \sum_{i \neq j} \texttt{edge}_T(z_t^i, z_t^j, a_t^i)) \tag{5}$$

## 4.2 REWARD MODEL

The reward model uses almost the same architecture as the transition model. Still, we average object embeddings returned by the node models and feed the result into the MLP to produce the scalar reward. The reward model is trained using the mean squared error loss function with environmental rewards $r_t$ as target (6).

$$\begin{cases} \texttt{embed}_R^i = \texttt{node}_R(z_t^i, a_t^i, \sum_{i \neq j} \texttt{edge}_R(z_t^i, z_t^j, a_t^i)) \\ \hat{R}(z_t, a_t) = MLP(\sum_{i=1}^K \texttt{embed}_R^i / K) \end{cases} \tag{6}$$

## 4.3 STATE-VALUE MODEL

The state-value function is approximated using a graph neural network $\hat{V}$, which does not depend on actions in either the edge model $\texttt{edge}_V$ or the node model $\texttt{node}_V$. As in the reward model, we average object embeddings returned by the node models and feed the result into the MLP to produce the scalar value.

$$\begin{cases} \texttt{embed}_V^i = \texttt{node}_V(z_t^i, \sum_{i \neq j} \texttt{edge}_V(z_t^i, z_t^j)) \\ \hat{V}(z_t) = MLP(\sum_{i=1}^K \texttt{embed}_V^i / K) \end{cases} \tag{7}$$

## 4.4 ACTOR MODEL

The actor model uses the same GNN architecture as the state-value model but employs different MLP heads for continuous and discrete action spaces. In the case of the continuous action space, it returns the mean and the covariance of the Gaussian distribution. For the discrete action space, it outputs the probabilities for all actions.

$$\begin{cases} \texttt{embed}_{actor}^i = \texttt{node}_{actor}(z_t^i, \sum_{i \neq j} \texttt{edge}_{actor}(z_t^i, z_t^j)) \\ \mu(z_t) = MLP_\mu(\sum_{i=1}^K \texttt{embed}_{actor}^i / K) \\ \sigma^2(z_t) = MLP_{\sigma^2}(\sum_{i=1}^K \texttt{embed}_{actor}^i / K) \\ \pi(z_t) = MLP_\pi(\sum_{i=1}^K \texttt{embed}_{actor}^i / K) \end{cases} \tag{8}$$

## 4.5 CRITIC MODEL

In the critic, we use a world model to predict action-values. Specifically, we employ a Q-function decomposition based on the Bellman equation. It was initially introduced in the Q-learning TreeQN algorithm (Farquhar et al., 2018):

$$\hat{Q}(z_t, a_t) = \hat{R}(z_t, a_t) + \gamma \hat{V}(z_t + \Delta z) \tag{9}$$

where $\hat{R}$ — the reward model 6, $\hat{V}$ — the state-value model 7, $\boldsymbol{z}_t + \Delta\boldsymbol{z}$ — the next state prediction, generated by the transition model 5. Since the critic's output values are computed using the world model, we refer to our approach as a value-based model-based method.

## 4.6 TRAINING

The SLATE model is pre-trained on the data set of trajectories collected with a uniform random policy (100K observations for Shapes2D tasks and 200K observations for the Object Reaching task). Following the original paper (Singh et al., 2022), we apply decay on the dVAE temperature $\tau$ from 1.0 to 0.1 and a learning rate warm-up for the parameters of the slot-attention encoder and the transformer at the start of the training. After pre-training, we keep the parameters of the SLATE model frozen.

To train all the other components of ROCA we use SAC objectives (2, 3, 4). For both continuous and discrete environments, a conventional double Q-network architecture is used in the critic module. Additionally, we use the data sampled from the replay buffer to train the world model components. The transition model is trained using the mean squared error loss function to minimize the prediction error of the object representations for the next state, given the action. The reward model is trained using the mean squared error loss function with environmental rewards $r_t$ as targets.

$$ J_{WM} = \mathbb{E}_{s_t,a_t,r_t,s_{t+1}\sim D}\left[\beta_T\|\boldsymbol{z}_t + \Delta\boldsymbol{z} - \boldsymbol{z}_{t+1}\|^2 + \beta_R\left(\hat{R}(\boldsymbol{z}_t, a_t) - r_t\right)^2\right] \quad (10) $$

In total, we use four optimizers. The temperature parameter, the actor, and the value model use individual optimizers. The transition and reward models share the world model optimizer.

Due to the stochastic nature of the SLATE model, object-centric representation can shuffle at each step. To enforce the order of object representation during the world model objective (10) optimization, we pre-initialize the slots of the SLATE model for the next state $\boldsymbol{z}_{t+1}$ with the current values $\boldsymbol{z}_t$.

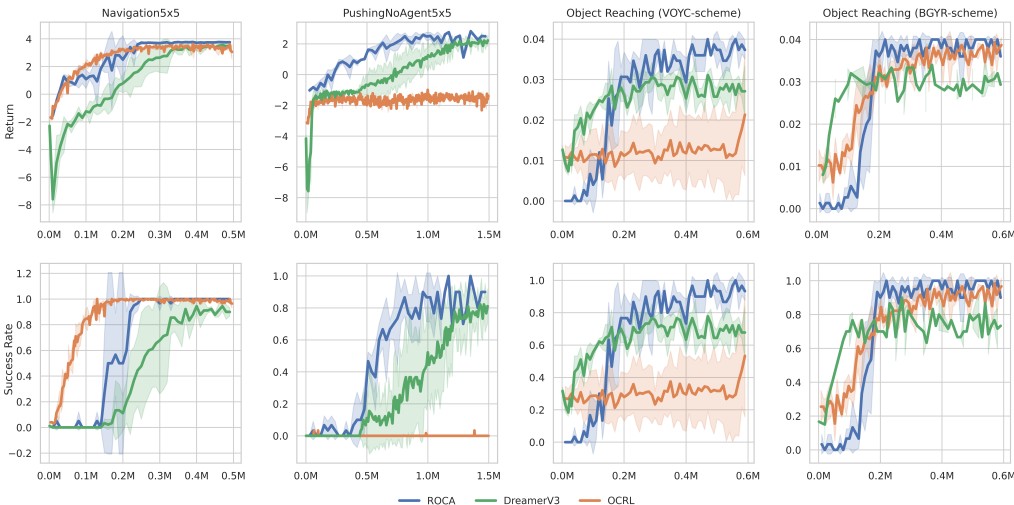

Figure 4: Return and success rate averaged over 30 episodes and three seeds for ROCA, DreamerV3, and OCRL models. ROCA learns faster or achieves higher metrics than the baselines. Shaded areas indicate standard deviation.

## 5 ENVIRONMENTS

The efficiency of the proposed ROCA algorithm was evaluated in the 3D robotic simulation environment CausalWorld (Ahmed et al., 2020) on the Object Reaching task as it was done in (Yoon et al., 2023), and in the compositional 2D environment Shapes2D (Kipf et al., 2020) on the Navigation and PushingNoAgent tasks. Current state-of-the-art slot-based object-centric models struggle

to extract meaningful object-centric representations in visually complex environments (Locatello et al., 2020a; Engelcke et al., 2021). As a result, testing object-centric RL algorithms in visually rich environments, like Habitat (Szot et al., 2022), becomes challenging due to the low quality of representations. However, the visual complexity of the selected environments enables object-centric models to extract high-quality object representations. This allows us to focus on the problem of object-centric MBRL, which is the primary objective of this paper.

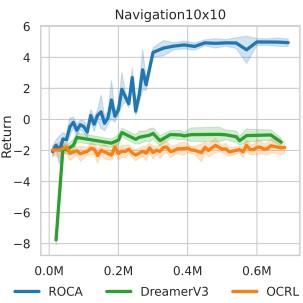

Figure 5: Return averaged over 30 episodes and three seeds for ROCA, DreamerV3, and OCRL models for Navigation 10x10 task. ROCA exhibits better performance than baselines but still does not solve the task. Shaded areas indicate standard deviation.

**Object Reaching Task** In this task, a fixed target object (violet cube) and a set of distractor objects (orange, yellow, and cyan cubes) are randomly placed in the scene. The agent controls a trifinger robot and must reach the target object with one of its fingers (the other two are permanently fixed) to obtain a positive reward and solve the task. The episode ends without reward if the finger first touches one of the distractor objects. The action space in this environment consists of the three continuous joint positions of the moveable finger. During our experiments, we discovered that one of the baseline algorithms is sensitive to the choice of color scheme for the cubes. Therefore, we also conducted experiments in the task with the original color scheme (Yoon et al., 2023): the color of the target cube is blue, and the colors of the distracting cubes are red, yellow, and green. Examples of observations are shown in appendix C.

**Navigation Task** Shapes2D environment is a four-connected grid world where objects are represented as figures of simple shapes. Examples of observations in the considered versions of the Shapes2D environment are shown appendix C. One object — the cross is selected as a stationary target. The other objects are movable. The agent controls all movable objects. In one step, the agent can move an object to any free adjacent cell. The agent aims to collide the controlled objects with the target object. Upon collision, the object disappears, and the agent receives a reward of $+1$. When an object collides with another movable object or field boundaries, the agent receives a reward of $-0.1$, and the positions of objects are not changed. For each step in the environment, the agent receives a reward of $-0.01$. The episode ends if only the target object remains on the field. In the experiments, we use a 5x5-sized environment with five objects and a 10x10-sized environment with eight objects. The action space in the Shapes2D environment is discrete and consists of 16 actions for the Navigation 5x5 task (four movable objects) and 28 actions for the Navigation 10x10 task (seven movable objects).

**PushingNoAgent Task** The agent controls all movable objects as in the Navigation task, but collisions between two movable objects are permitted: both objects move in the direction of motion. The agent is tasked to push another movable object into the target while controlling the current object. The pushed object disappears, and the agent receives a reward of $+1$ for such an action. When the currently controlled object collides with the target object or field boundaries, the agent receives a reward of $-0.1$. When the agent pushes a movable object into the field boundaries, the agent receives a reward of $-0.1$. For each step in the environment, the agent receives a reward of $-0.01$. The episode ends if only the target object and one movable object remain on the field. In the experiments, we use a 5x5-sized environment with five objects.

## 6 EXPERIMENTS

We utilize a single SLATE model for Navigation5x5 and PushingNoAgent5x5 tasks as they share the same observation space. However, we train a distinct SLATE model for Navigation10x10 and each version of the Object Reaching task. Appendix A provides detailed information regarding the hyperparameters of the SLATE model.

In continuous Object Reaching tasks, we conventionally use the dimension of the action space as the target entropy hyperparameter for ROCA. For 2D tasks with a discrete action space, we scale the entropy of a uniform random policy with the tuned coefficient. For more information on the hyperparameters of the ROCA model, please refer to appendix A.

We compare ROCA with a model-free algorithm based on PPO, using the same pre-trained frozen SLATE model as a feature extractor. To combine the latent object representations into a single vector suitable for the value and policy networks of the PPO, we used a Transformer encoder (Vaswani et al., 2023) as a pooling layer. We referred to the transformer-based PPO implementation provided by (Yoon et al., 2023) as the OCRL baseline. For the Object Reaching Task, we employed the same hyperparameter values as the authors. For Shapes2D tasks, we fine-tuned the hyperparameters of the OCRL baseline. The tested values are listed in the appendix D. Since there are no established state-of-the-art object-centric MBRL algorithms, we have chosen the DreamerV3 (Hafner et al., 2023) algorithm as a MBRL baseline. In order to ensure a fair comparison between the ROCA and the DreamerV3, we conducted experiments where we trained the DreamerV3 with a pretrained encoder obtained from the DreamerV3 model that solves the task. For all the tasks, we conducted experiments using two different modes: one with the encoder frozen and another with the encoder unfrozen. However, we did not observe any improvement in the convergence rate compared to the DreamerV3 model that does not use the pretrained encoder. Additionally, we discovered that the pretrained world model significantly accelerates the convergence of DreamerV3, but this mode makes the comparison unfair to the ROCA. For the DreamerV3 algorithm we use default hyperparameter values from the official repository. The results of an additional experiment evaluating out-of-distribution generalization to unseen colors in the Object Reaching task can be found in appendix F.

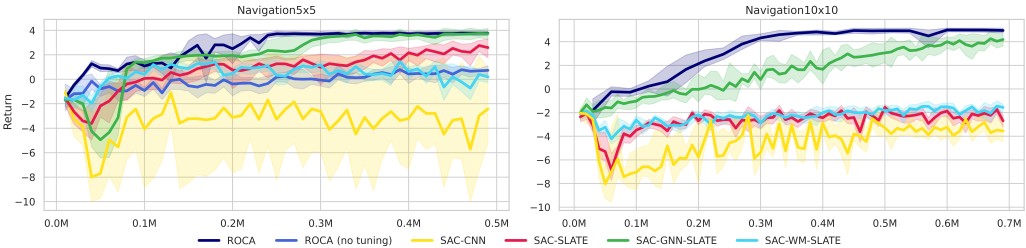

Figure 6: Ablation study. SAC-CNN — a version of SAC with a standard CNN encoder. SAC-SLATE — a version of SAC with a pretrained SLATE encoder which averages object emebeddings to obtain the embedding of the current state. SAC-WM-SLATE — a modification of SAC-SLATE which uses a monolithic world-model in its critic. SAC-GNN-SLATE — an object-centric version of SAC with a pretrained SLATE encoder which uses GNNs as actor and critic. ROCA (no-tuning) — a version of ROCA without target entropy tuning. ROCA outperforms the considered baselines. Shaded areas indicate standard deviation.

**Results**  The graphs in Figure 4 depict how the episode return of ROCA and the baselines depend on the number of steps for Navigation 5x5, PushingNoAgent5x5, and two versions of the Object Reaching task. For the Navigation 5x5 task, ROCA performs better than the OCRL baseline. Although DreamerV3 shows slightly more stable and efficient learning than ROCA, ROCA eventually achieves a higher return. In the PushingNoAgent 5x5 task, ROCA outperforms both baselines. The baselines are initially more effective in the Object Reaching task with our color scheme, but ROCA outperforms them after 200K steps. For the Object Reaching task with the original color scheme, the OCRL baseline demonstrates much better performance, but ROCA also surpasses both baselines after 200K steps. Figure 5 demonstrates the results in the more challenging Navigation 10x10 task. Both baselines fail to achieve a positive return. ROCA performs better than both baselines but can not solve the task entirely, as it only moves five out of seven objects to the target. We believe that the poor performance of the OCRL baseline in the Object Reaching task with VOYC color schema is due to its sensitivity to the quality of the SLATE model. One potential solution to overcome this issue could be increasing the number of training epochs for the SLATE.

**Ablations**   ROCA is built upon SAC, and thus, the ablation study aims to assess the impact of the different modifications we introduced to the original SAC with a monolithic CNN encoder. Figure 6 illustrates the results of additional experiments estimating the effects of the pre-trained SLATE encoder, the object-centric actor and critic, the object-centric world model and the target entropy tuning. We evaluate the quality of several monolithic and object-centric versions of SAC and compare them with ROCA. SAC-CNN is standard monolithic version of SAC that utilizes the convolutional encoder from the original DQN implementation (Mnih et al., 2015). In SAC-SLATE, the CNN encoder is replaced with a pre-trained frozen SLATE encoder, while the other model components remain the same. To obtain the monolithic state representation $z_t^*$ from the object-centric one $z_t$, produced by the SLATE, we take the average over the object axis: $z_t^* = \sum_{i=0}^{K} z_t^i / K$. Note, that $z_t^*$ is independent of the slot order in $z_t$ and can be fed into the standard actor and critic MLPs. SAC-WM-SLATE builds upon SAC-SLATE and can be considered as a monolithic version of the ROCA. Its actor, state-value, reward, and transition models are implemented using MLPs. SAC-GNN-SLATE is an object-centric version of SAC and can be viewed as ROCA without the world model in the critic module. It uses a pretrained frozen SLATE encoder and GNN-based actor and critic modules. Additionally, we compare the ROCA with a variant where the target entropy is set to the default value, equal to the scaled entropy of the uniform random policy with coefficient 0.98 (Christodoulou, 2019).

The ablation studies have shown that in the monolithic mode, the SLATE model significantly improves performance only in the relatively simple Navigation5x5 task. However, extending the critic with the world model does not improve the convergence rate. The object-centric SAC-GNN-SLATE outperforms all monolithic models. Finally, the ROCA, which uses an object-centric world model in the critic module, outperforms the SAC-GNN-SLATE. Note that we obtained the presented results after fine-tuning the hyperparameters for all of the models.

## 7   CONCLUSION AND FUTURE WORK

We presented ROCA, an object-centric off-policy value-based model-based reinforcement learning approach that uses a pre-trained SLATE model as an object-centric feature extractor. Our experiments in 3D and 2D tasks demonstrate that ROCA learns effective policies and outperforms object-centric model-free and model-based baselines. The world model is built upon a GNN architecture, showing that graph neural networks can be successfully applied in MBRL settings for policy learning. While we use the SLATE model as an object-centric feature extractor, in principle, we can replace SLATE with other slot-based object-centric models. However, ROCA does have limitations. Firstly, its world model is deterministic and may struggle to predict the dynamics of highly stochastic environments. Additionally, as our model is based on the SAC algorithm, it is sensitive to the target entropy hyperparameter, especially in environments with discrete action spaces (Xu et al., 2021; Zhou et al., 2023).

In our future work, we consider the primary task to be evaluating the ROCA in more visually challenging environments. To accomplish this, we plan to replace the SLATE with the recently proposed DINOSAUR (Seitzer et al., 2023) model, which has shown promising results on realistic datasets. In addition, we have plans to experiment with non-slot object-centric approaches, such as Deep Learning Particles (Daniel & Tamar, 2022). Our plans include enhancing the model's robustness to changes in the target entropy by adopting a metagradient-based approach (Wang & Ni, 2020), which eliminates the need for this hyperparameter. In many environments, only a small number of objects interact at a time. Therefore, representing the environment's state as a complete graph leads to redundant connections. To address this issue, we plan to implement approaches (Goyal et al., 2022; Zadaianchuk et al., 2022) that sparsify the state graph.

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

## A    IMPLEMENTATION DETAILS

The edge and the node model in GNN-based transition model, reward model, value model and actor model are MLP's which consists of two hidden layers of 512 units each, LayerNorm and ReLU activations.

Target entropy parameter is set to -3 for Object Reaching task. For tasks with discrete action space we use the scale the entropy of a uniform random policy with coefficient 0.6, which means 1.66 for Navigation 5x5 and PushingNoAgent 5x5 tasks and 2 for Navigation 10x10 task.

| | | |
|---|---|---|
| Learning | Temp. Cooldown | 1 |
| | Temp. Cooldown Steps | 30000 |
| | LR for DVAE | 0.0003 |
| | LR for CNN Encoder | 0.0001 |
| | LR for Transformer Decoder | 0.0003 |
| | LR Warm Up Steps | 30000 |
| | LR Half Time | 250000 |
| | Dropout | 0.1 |
| | Clip | 0.05 |
| | Batch Size | 24 |
| | Epochs | 150 |
| DVAE | vocabulary size | 4096 |
| CNN Encoder | Hidden Size | 64 |
| Slot Attention | Iterations | 3 |
| | Slot Heads | 1 |
| | Slot Dim. | 192 |
| | MLP Hidden Dim. | 192 |
| | Pos Channels | 4 |
| Transformer Decoder | Layers | 4 |
| | Heads | 4 |
| | Hidden Dim | 192 |

Table 1: Hyperparameters for SLATE

| | |
|---|---|
| Gamma | 0.99 |
| Buffer size | 1000000 |
| Batch size | 128 |
| $\tau_{polyak}$ | 0.005 |
| Buffer prefill size | 5000 |
| Number of parallel environments | 16 |

Table 2: Hyperparameters for ROCA

## B    SOFT ACTOR CRITIC

### B.1    SOFT ACTOR-CRITIC FOR CONTINUOUS ACTION SPACES

Soft Actor-Critic (SAC) (Haarnoja et al., 2018; 2019) is a state-of-the-art off-policy reinforcement learning algorithm for continuous action settings. The goal of the algorithm is to find a policy that maximizes the maximum entropy objective:

$$\pi^* = \mathrm{argmax}_\pi \sum_{i=0}^{\tau} \mathbb{E}_{(s_t, a_t) \sim d_\pi} \left[ \gamma^t (R(s_t, a_t) + \alpha \mathcal{H}(\pi(\cdot | s_t))) \right]$$

where $\alpha$ is the temperature parameter, $\mathcal{H}(\pi(\cdot|s_t)) = -\log\pi(\cdot|s_t)$ is the entropy of the policy $\pi$ at state $s_t$, $d_\pi$ is the distribution of trajectories induced by policy $\pi$. The relationship between the soft state-value function and the soft action-value function is determined as

$$V(s_t) = \mathbb{E}_{a_t\sim\pi(\cdot|s_t)}[Q(s_t, a_t) - \alpha\log(\pi(a_t|s_t))] \tag{11}$$

The soft action-value function $Q_\theta(s_t, a_t)$ parameterized using a neural network with parameters $\theta$ is trained by minimizing the soft Bellman residual:

$$J_Q(\theta) = \mathbb{E}_{(s_t,a_t)\sim D}\big[\big(Q_\theta(s_t, a_t) - R(s_t, a_t) - \gamma\mathbb{E}_{s_{t+1}\sim T(s_t,a_t)}V_{\bar{\theta}}(s_{t+1})\big)^2\big] \tag{12}$$

where $D$ is a replay buffer of past experience and $V_{\bar{\theta}}(s_{t+1})$ is estimated using a target network for $Q$ and a Monte Carlo estimate of (11) after sampling experiences from the $D$.

The policy $\pi$ is restricted to a tractable parameterized family of distributions. A Gaussian policy is often parameterized using a neural network with parameters $\phi$ that outputs a mean and covariance. The parameters are learned by minimizing the expected KL-divergence between the policy and the exponential of the $Q$-function:

$$J_\pi(\phi) = \mathbb{E}_{s_t\sim D}\big[\mathbb{E}_{a_t\sim\pi_\phi(\cdot|s_t)}\big[\alpha\log(\pi_\phi(a_t|s_t)) - Q_\theta(s_t, a_t)\big]\big] \tag{13}$$

After reparameterization of the policy with the standard normal distribution, the (13) becomes feasible for backpropagation:

$$J_\pi(\phi) = \mathbb{E}_{s_t\sim D,\epsilon_t\sim\mathcal{N}(0,1)}\big[\alpha\log(\pi_\phi(f_\phi(\epsilon_t; s_t)|s_t)) - Q_\theta(s_t, f_\phi(\epsilon_t; s_t))\big] \tag{14}$$

where action are parameterized as $a_t = f_\phi(\epsilon_\phi; s_t)$.

The objective for the temperature parameter is given by:

$$J(\alpha) = \mathbb{E}_{a_t\sim\pi(\cdot|s_t)}\big[-\alpha(\log\pi(a_t|s_t) + \bar{H})\big] \tag{15}$$

where $\bar{H}$ is a hyperparameter representing the target entropy. In practice, two separately trained soft Q-networks are maintained, and then the minimum of their two outputs are used to be the soft Q-network output.

## B.2 SOFT ACTOR-CRITIC FOR DISCRETE ACTION SPACES

While SAC solves problems with continuous action space, it cannot be straightforwardly applied to discrete domains since it relies on the reparameterization of Gaussian policies to sample action. A direct discretization of the continuous action output and Q value (SACD) was suggested by (Christodoulou, 2019). In the case of discrete action space, $\pi_\phi(a_t|s_t)$ outputs a probability for all actions instead of a density. Thus, the expectation (11) can be calculated directly and used in the Q-function objective (12):

$$V(s_t) = \pi(s_t)^T\big[Q(s_t) - \alpha\log\pi(s_t)\big] \tag{16}$$

The temperature objective (15) changes to:

$$J(\alpha) = \pi(s_t)^T\big[-\alpha(\log\pi(s_t) + \bar{H})\big] \tag{17}$$

The expectation over actions in (13) can be calculated directly, which leads to the policy objective:

$$J_\pi(\phi) = \mathbb{E}_{s_t\sim D}\big[\pi(s_t)^T\big[\alpha\log(\pi_\phi(s_t)) - Q_\theta(s_t, \cdot)\big]\big] \tag{18}$$

## C SLATE

The SLATE (Singh et al., 2022) model is used as an object-centric representations extractor from image-based observations $s_t$. It consists of a slot-attention module (Locatello et al., 2020c), dVAE, and GPT-like transformer (Ramesh et al., 2021).

The purpose of dVAE is to reduce an input image of size $H\times W$ into lower dimension representation by a factor of $K$. First, the observation $s_t$ is fed into the encoder network $f_\phi$, resulting in log

probabilities $o_t$) for a categorical distribution with $C$ classes. Then, these log probabilities are used to sample relaxed one-hot vectors $j_t^{\text{soft}}$ from the relaxed categorical distribution with temperature $\tau$. Each token from $j_t^{\text{soft}}$ represents information about $K \times K$ size patch of overall $P = HW/K^2$ patches on the image. After that, $j_t^{\text{soft}}$ the vector is being used to reconstruct observation $\tilde{s}_t$ by these patches with the decoder network $g_\theta$.

$$\begin{cases} o_t = f_\phi(s_t) \\ j_t^{\text{soft}} \sim \text{RelaxedCategorical}(o_t; \tau); \\ \tilde{s}_t = g_\theta(j_t^{\text{soft}}). \end{cases}$$

The training objective of dVAE is to minimize MSE between observation $s_t$ and reconstruction $\tilde{s}_t$:

$$L_{dVAE} = \text{MSE}(\mathbf{s_t}, \tilde{\mathbf{s}}_\mathbf{t}), \tag{19}$$

Discrete tokens $j_t$, obtained from categorical distribution, are mapped to embedding from learnable dictionaries. Those embeddings are summed with learned position embedding $p_\phi$ to fuse information about patches on the image. Then, the resulting embeddings $u_t^{1:P}$ are fed into the slot attention module. The slot attention returns $N$ object slots $z_t^{1:N}$, which are vectors of the fixed dimension Slot Dim, along with $N$ attention maps $A_t^{1:N}$.

$$\begin{cases} o_t = f_\phi(s_t); \\ j_t \sim \text{Categorical}(o_t); \\ u_t^{1:P} = \text{Dictionary}_\phi(j_t) + p_\phi; \\ z_t^{1:N}, A_t^{1:N} = \text{SlotAttention}_\phi(u_t^{1:P}). \end{cases}$$

The transformer predicts log-probabilities autoregressively $\hat{o}_t^i$ for path $i$ from vectors $\hat{u}_t^{<i}$ generated for previous patches, combined with object centric representations $z_t^{1:N}$. The vector $\hat{u}_t^l$, $l < i \in [1 : P]$ is formed dictionary embedding from previously generated token $\hat{j}_t^l$ for path $l$ with added position embedding $p_\phi^l$. The token $\hat{j}_t^i$ is mapped to the class $c \in C$ with the highest log-probability $\hat{o}_{t,c}^i$. The resulting token can be used to reconstruct observation $\hat{s}_t$ by combining reconstructed patches $\hat{s}_t^i$.

$$\begin{cases} \hat{u}_t^{<i} = \text{Dictionary}_\phi(\hat{j}_t^{<i}) + p_\phi^i; \\ \hat{o}_t^i = \text{Transformer}_\theta(\hat{u}_t^{<i}; z_t^{1:N}); \\ \hat{j}_t^i = \arg\max_{c \in [1,C]} \hat{o}_{t,c}^i; \\ \hat{s}_t^i = g_\theta(\hat{j}_t^i). \end{cases}$$

The training objective of the transformer is to minimize cross entropy between the distribution of tokens $\hat{j}_t$ generated by the transformer and tokens $j_t$ extracted by dVAE.

$$L_{\text{T}} = \sum_{i=1}^{P} \text{CrossEntropy}(z_t^i, \hat{z}_t^i) \tag{20}$$

Combining 19 and 20 we receive loss for the SLATE model:

$$L_{SLATE} = L_{dVAE} + L_T$$

Figure 7 illustrates the examples of original observations and slot-attention masks learned by the SLATE model in the Object Reaching and Shapes2D tasks.

## D  OCRL FINE-TUNING DETAILS

We conducted additional experiments to tune the OCRL baseline in the tasks where it was outperformed by ROCA. In Navigation10x10 and PushingNoAgent5x5 tasks we went through combinations of hyperparameters, but did not observe significant improvements:

- Entropy coefficient: [0, 0.001, 0.01, 0.025, 0.05, 0.075, 0.1]
- Clip range: [0.1, 0.2, 0.4]
- Epochs: [10, 20, 30]
- Batch size: [64, 128]

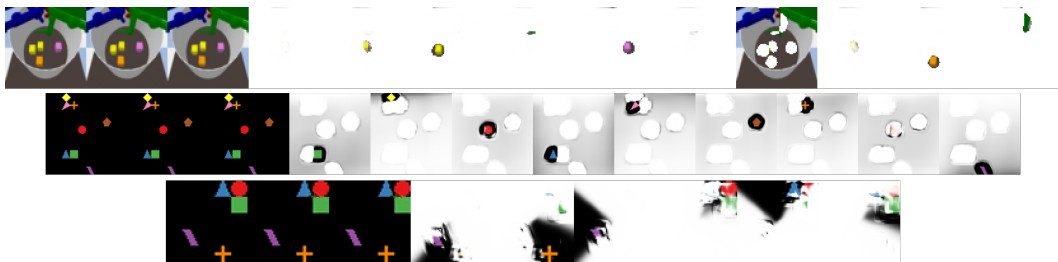

Figure 7: Examples of observations and slots extracted by the SLATE model in the Object Reaching task (top), Navigation 10x10 task (middle), and Navigation 5x5 task (bottom).

# E    ADDITIONAL EXPERIMENTS WITH DREAMERV3

In order to ensure a fair comparison with DreamerV3, we conducted experiments with a pretrained encoder obtained from the DreamerV3 model that solves the task. For all the tasks, we conducted experiments using two different modes: one with the encoder frozen and another with the encoder unfrozen. However, we did not observe any improvement in the convergence rate. The results are shown in the Figure 8.

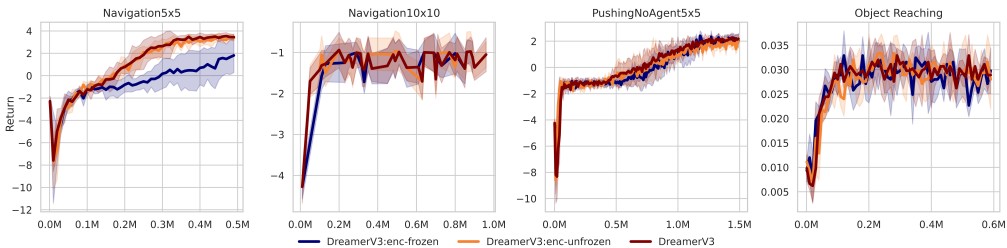

Figure 8: The plots illustrate the impact of encoder pretraining for DreamerV3 algorithm. *DreamerV3* is a default version that trains its encoder from scratch. *DreamerV3:enc-fronzen* is a version with a pretrained frozen encoder. *DreamerV3:enc-unfrozen* is a version with a pretrained unfrozen encoder. Return and success rate averaged over 30 episodes and three seeds for different.

# F    EVALUATION OF OUT-OF-DISTRIBUTION GENERALIZATION TO UNSEEN COLORS

We evaluated the generalization of the ROCA to unseen colors of distractor objects in the Object Reaching task. When we tested the model with the same colors it was trained on, it achieved a success rate of $0.975 \pm 0.005$. However, when we used new colors for the distractor objects, the success rate dropped to $0.850 \pm 0.005$. The results were averaged over three instances of the ROCA models, and each model was evaluated on 100 episodes.

