# OpenReview forum: "Graphical Object-Centric Actor-Critic"
_ICLR.cc/2024/Conference — Submitted to ICLR 2024_

### Official Review · Reviewer_rSLu · 2023-10-27

**Soundness:** 2 fair
**Presentation:** 2 fair
**Contribution:** 2 fair
**Rating:** 5
**Confidence:** 4

**Summary:**

This paper proposes an object-centric actor-critic algorithm, where first object-centric representations are extracted from pixels using SLATE, and then a GNN is used to train the policy and Q-value prediction. The method was evaluated on object reaching, pushing, and navigation tasks, and compared with Dreamerv3 and a transformer-based OCRL baseline.

**Strengths:**

- The paper combines a number of elements (SAC, SLATE, GNN transition model) and demonstrates that it works.

**Weaknesses:**

- The method claims to outperform the baselines, but it feels like the particular environments are somehow crafted / cherry-picked / designed to have some failure cases for the baselines. i.e. it is striking that for Navigation and Object reaching the OCRL method seems on par or even better, but it completely fails on this PushingNoAgent and Object reaching with different color scheme environment, which are less standard benchmarks.

- It is unclear from the paper and method description exactly which parts are novel for this paper. i.e. 4.1 simply rehearses SLATE and 4.2-4.5 utilize the GNN architecture as described by Kipf et al 2020? I assume that the main novelty is in the combination + training end-to-end with SAC?

**Questions:**

- Any idea on why the OCRL method fails on particularly the PushingNoAgent and Object Reaching with the VOYC color scheme tasks, but is on par or even better on the others? Also the fact that the OCLR method completely fails on a task seems like rather a bug or wrongly tuned hyperparameters, since there shouldn't be any significant difference in the problem structure...

- As a model-based baseline DreamerV3 has no structured latent space as opposed to the proposed method. An interesting comparison would be to have a model-based baseline that uses structured world models, e.g. like the one proposed in Kipf et al.

---

> ### Author Response · Authors · 2023-11-20
>
> Thank you for your valuable feedback. Please find our responses to your questions below.
>
> *[Performance of the OCRL baseline]*
> 1. The GOCA and the OCRL baseline both use the same pretrained frozen SLATE model as an object-centric encoder. However, in the case of the VOYC color schema, on average, the OCRL baseline converges in only one out of three runs. We hypothesise that the transformer-based OCRL is more sensitive to the quality of the SLATE model. To potentially address this issue, increasing the number of training epochs for the SLATE on the VOYC dataset could be considered.
> 2. We attempted to tune the OCRL baseline in the tasks where it was outperformed by GOCA. We went through combinations of hyperparameters, but did not observe significant improvements: entorpy_coefficient=[0, 0.001, 0.01, 0.025, 0.05, 0.075, 0.1], clip_range=[0.1, 0.2, 0.4], epochs=[10, 20, 30], batch_size=[64, 128].
>
> *[DreamerV3]* \
> We chose DreamerV3 to demonstrate that even a powerful monolithic model-based algorithm could be ourperformed by a simpler model-based algorithm such as GOCA, which utilizes the object-centric structure of the environment. We believe that an object-centric extension of DreamerV3 could potentially achieve better performance in the considered environments. However, to the best of our knowledge, no one has proposed such an extension yet. In fact, we have plans to develop an object-centric version of DreamerV3 ourselves. As there is no established SOTA for object-centric MBRL, we would greatly appreciate any suggestions regarding which algorithms we should use as baselines for our method.
>
>
> Our responses to the highlighted weaknesses:
>
> *[Environments]*
> 1. Current SOTA slot-based object-centric models struggle in extracting meaningful object-centric representations in visually complex environments. Testing our RL approach in visually rich environments, such as Habitat [1] (proposed by Reviewer 6gyQ), becomes challenging because of low quality of representations. Visual complexity of the selected environments allows object-centric models extract good quality object representations. While our 2D environments adhere to standard object-centric RL tasks, they present more challenging dynamics compared to the environments where the transformer-based OCRL baseline was initially evaluated. For instance, in Navigation5x5 and PushingNoAgent5x5, there are four movable objects, whereas in Navigation10x10, there are seven movable objects. In contrast, the 2D environments [2] typically have one or two movable objects.
> 2. Both the GOCA and the OCRL baseline use the same pretrained frozen SLATE model as an object-centric encoder. However, the OCRL baseline, on average, converges in one out of three runs for the VOYC color schema. We believe that the transformer-based OCRL is more sensitive to the quality of the SLATE model, and this issue can potentially be overcome by increasing the number of training epochs for the SLATE on VOYC dataset. \
> [1] Szot el al. Habitat 2.0: Training Home Assistants to Rearrange their Habitat. 2021 \
> [2] Jaesik Yoon et al. An Investigation into Pre-Training Object-Centric Representations for Reinforcement Learning, 2023
>
> *[Novelty]* \
> We will explicitly highlight the contribution here and in the revised version of the paper:
> 1) We proposed a novel architecture that combines a value-based model-based approach with the actor-critic method.
> 2) We extended the SAC algorithm by introducing a new objective function to train the model-based critic.
> 3) We proposed using a GNN-based actor to pool object-centric representations.
> 4) We modified the CSWM transition model by adjusting its edge model: we pass a pair of slots into the edge model along with an action.

---

> > ### Comment · Reviewer_rSLu · 2023-11-22
> >
> > Thank you for the response. I acknowledge that indeed there is no off-the-shelf object-centric, model-based algorithm readily available (i.e. an object-centric Dreamer). I appreciate the four explicit contributions stated in the comment, and the paper would vastly improve if it was more centered around these contributions. I would also expect these points to be addressed in the ablation study, i.e. can you quantify the effect of adjusting the CSWM edge model?
> >
> > However, given the current experimental results, it's still unclear how significant these improvements are. Considering also the other reviewer's feedback and discussion, I'm keeping my borderline score.

---

> ### Author Response · Authors · 2023-11-23
>
> We would like to thank you for your helpful critical feedback.
> We have updated our manuscript based on your feedback:
> 2) We restated our contribution in the introduction
> 3) We addressed your question about OCRL pretraining in the appendix.
> 3) We addressed the questions that were raised during the rebuttal phase.
> 4) The updated parts are highlighted.
>
> Unfortunatelly we did not have enough time to conduct the experiments comparing original CSWM edge model and our version. We will include the results in the final version.

---

### Official Review · Reviewer_aEan · 2023-10-29

**Soundness:** 2 fair
**Presentation:** 2 fair
**Contribution:** 1 poor
**Rating:** 3
**Confidence:** 4

**Summary:**

The paper proposes an RL method based on object-centric representations and world models. The object-centric representation is based on SLATE, the components built on top of the representation (transition, reward, value and actor model) use graph neural networks and the RL algorithm is based on SAC (continuous and discrete). The work is evaluated on 5 variations of 2 environments.

**Strengths:**

* **Ideas**: there are several promising ideas combined together in this work, such as the idea of using object-centric representations together with world models, and the idea of using graph neural networks for the transition dynamics.

**Weaknesses:**

* **Model-based approach?**: it is not clear to me how this is a model-based approach for RL. Are the world model components (ie. transition and reward models) only used to improve the representation? Or is the actor-critic actually learned in imagination and this is not detailed in the paper?
* **Novelty**: the work combines several ideas from the literature (SLATE, the GNN from CSWM, SAC in continuous and discrete settings) into one method. While it is interesting to see that this works, in the tested environments, a strong novel contribution seems to be missing from the paper.
* **Evaluation**: the experiments in the paper are mainly limited to two tasks/environments (with variations). It is unclear what the authors are trying to show, as they mention `efficiency' but in the standard versions of the tasks the "OCRL" baseline seems to be more efficient, as it converges faster in the 5x5 Navigation task and it starts learning faster and converges to a similar result in the BGYR Object Reaching task. I also find the ablation study unclear: if the approach is model-based, how are SAC-based approaches ablations? How does the tuning of the entropy term cause such a large difference in performance?

**Questions:**

* Why is the OCRL baseline affected by the changes in colors and GOCA is not? Aren't they both based on SLATE for the encoding?
* I think the evaluation is unfair to DreamerV3 as this baseline's representation is not pre-trained on any data from the environment, while GOCA (and possibly OCRL too?) representation is pre-trained. I would ask the authors to find a way to make this comparison fair (e.g. pre-training the DreamerV3 world model too)
* How many random steps from the environment are used for pretraining SLATE?

Typos/writing suggestions:
* In Section 4.2, "We approximate THE transition function" --> (missing THE)
* In the Conclusion, "We presented GOCA, an object-centric off-policy ~~value-based~~ " -> I wouldn't call the approach value-based as it uses an actor-critic

---

> ### Author Response · Authors · 2023-11-20
>
> Thank you for your feedback. Please see our responses to your questions. \
> *[Why is the OCRL baseline affected by the changes in colors and GOCA is not? Aren't they both based on SLATE for the encoding?]* \
> Both the GOCA and the OCRL baseline use the same pretrained frozen SLATE model as an object-centric encoder. However, the OCRL baseline, on average, converges in one out of three runs for the VOYC color schema. We believe that the transformer-based OCRL is more sensitive to the quality of the SLATE model, and this issue can potentially be overcome by increasing the number of training epochs for the SLATE on VOYC dataset.
>
> *[Comparison with DreamerV3]* \
> Please note that for the GOCA and the OCRL baseline, we only pretrain and freeze the object-centric encoder SLATE, not the world model. In order to ensure a fair comparison with DreamerV3, as you suggested, we conducted experiments where we trained DreamerV3 with a pretrained encoder obtained from the DreamerV3 model that solves the task. In all the tasks we considered, we tried both settings: where the encoder is frozen and where it is not frozen. However, we did not observe any improvement in the convergence rate. Additionally, we discovered that the pretrained world model significantly accelerates the convergence of DreamerV3, but this setting makes the comparison unfair to the GOCA.
>
> *[How many random steps from the environment are used for pretraining SLATE?]* \
> For the Shapes2D tasks, we pretrain the SLATE model using datasets consisting of 100K observations collected with a uniform random policy. As for the Object Reaching task, the dataset comprises 200K observations collected with a uniform random policy.

---

> ### Author Response · Authors · 2023-11-20
>
> Our responses to the highlighted weaknesses:
>
> *[Model-based approach?]*
> 1. We refer to our approach as a value-based **model-based** method, as we use a world model in the critic module to predict action-values. Namely, we employ decomposition: $ Q(s_t, a) = r_{WM}(s_t, a) + \gamma V(s_{t + 1}) \text{  and  } s_{t + 1} = T_{WM}(s_t, a) $.
>  Our approach is inspired by the Q-learning TreeQN [1] algorithm, which applies a world model and a state-value model to estimate Q-values and demonstrates superior performance compared to DQN.
> 2. Since we use the pre-trained frozen SLATE model to extract object-centric representations, the world model components do not affect representation learning, but the transition model is trained to predict representations for the next state. The next state representation prediction can be viewed as an imagination step in our approach. However, the actor and critic are trained using data from the replay buffer, which stores actual environment samples. \
> [1] Gregory Farquhar et al. TreeQN and ATreeC: Differentiable Tree-Structured Models for Deep Reinforcement Learning, 2018.
>
> *[Novelty]* \
> We will explicitly highlight the contribution here and in the revised version of the paper as it is not clear for readers:
> 1. We proposed a novel architecture that combines a value-based model-based approach with the actor-critic method.
> 2. We extended the SAC algorithm by introducing a new objective function to train the model-based critic.
> 3. We proposed using a GNN-based actor to pool object-centric representations.
> 4. We modified the CSWM transition model by adjusting its edge model: we pass a pair of slots into the edge model along with an action.
>
> *[Evaluation. Environments]*
> 1. Current SOTA slot-based object-centric models struggle in extracting meaningful object-centric representations in visually complex environments. Testing our RL approach in visually rich environments, such as Habitat [2] (proposed by Reviewer 6gyQ), becomes challenging because of low quality of representations. The recently introduced DINOSAUR [3] model seems suitable for visually rich environments, and we plan to conduct experiments with it. However, in this work, we focus on integrating slot-based representations and object-centric world model in a reinforcement learning algorithm, rather than on the improving of slot-based representation models themselves. Therefore, we have chosen to use standard for object-centric RL environments where current object-centric representation models perform well.
> 2. By "efficiency" we mean that GOCA performs on par with model-free algorithms in simple tasks like Navigation5x5. However, it scales better as the number of objects increases (Navigation10x10) or the environment dynamics becomes more challenging (PushingNoAgent5x5). \
> [2] Szot el al. Habitat 2.0: Training Home Assistants to Rearrange their Habitat. 2021 \
> [3] Seitzer el al. Bridging the Gap to Real-World Object-Centric Learning. 2023
>
> *[Evaluation. If the approach is model-based, how are SAC-based approaches ablations?]* \
> Our approach is built upon SAC, and thus, our ablation study aims to assess the impact of the different modifications we introduced to the original SAC (with a monolithic CNN encoder). Specifically, evaluated the contributions of the pretrained frozen SLATE encoder and the entropy tuning. Additionally, in the revised version of our paper, we will include a comparison with GNN-based SAC without a world model.
>
> *[Evaluation. How does the tuning of the entropy term cause such a large difference in performance?]* \
> It is a known issue that discrete SAC is sensitive to hyperparameters related to entropy [4, 5]. Since our approach is based on SAC so it also inherits this flaw. However, we consider this problem to be beyond the scope of our work. As part of our future plans, we intend to adopt a metagradient-based approach [6] that does not require the target entropy hyperparameter. \
> [4] Yaosheng Xu, et. al. Target Entropy Annealing for Discrete Soft Actor-Critic, 2021 \
> [5] Haibin Zhou, et. al. Revisiting Discrete Soft Actor-Critic, 2023 \
> [6] Yufei Wang, Tianwei Ni. Meta-SAC: Auto-tune the Entropy Temperature of Soft Actor-Critic via Metagradient. 2020
>
> *[Writing suggestions. I wouldn't call the approach value-based as it uses an actor-critic]* \
> We refer to our approach as a value-based model-free method, as we use a world model in the critic module to predict action-values.

---

> > ### Comment · Reviewer_aEan · 2023-11-22
> >
> > Dear authors,
> >
> > I thank you for addressing my questions. In general, I feel more positive about the paper, after reading your clarifications. However, there seem to be several parts that are missing from the original manuscript that I believe are important in order to understand the work better.
> >
> > More detailedly, referring to your comments, I believe you should:
> > - empirically verify your claim about the need to train the SLATE encoder for longer on the VOYC dataset
> > - add the experiments where Dreamer's encoder is also pretrained
> > - explain better why the method could be referred to as "value-based model-based" (the reference [1] you provided is missing from the manuscript and the Critic section is less than two lines at the moment)
> >
> > If you managed to revise the manuscript with the changes above, I would be happy to increase my score to reflect my more positive opinion towards the paper. However, I still feel a bit hesitant to recommend acceptance due to the limited evaluation, which doesn't really show a clear advantage in adopting GOCA compared to the baselines, even on these simple tasks.

---

> ### Author Response · Authors · 2023-11-23
>
> We would like to thank you for your helpful critical feedback.
> We have updated our manuscript based on your feedback:
> 1) We added to appendix plots of experiments with pretrained encoder for DreamerV3.
> 2) We restated our contribution in the introduction
> 3) Added explanation about value-based model-based approach in "Critic Model" section.) We addressed the questions that were raised during the rebuttal phase.
> 3) We addressed the questions that were raised during the rebuttal phase.
> 4) The updated parts are highlighted.
>
> Unfortunatelly we did not have enough time to conduct the experiment with VOYC dataset before the end of the rebuttal phase. We will include the results in the final version.

---

### Official Review · Reviewer_HvnL · 2023-11-01

**Soundness:** 2 fair
**Presentation:** 2 fair
**Contribution:** 2 fair
**Rating:** 6
**Confidence:** 3

**Summary:**

This manuscript integrates object-centric representation learning with object-relational dynamics, aiming to construct a dynamic model directly from pixel data for use in model-based reinforcement learning. On a conceptual and motivational level, the paper addresses a pivotal issue in RL, specifically in scenarios involving multiple objects or entities.

While the underlying idea is straightforward, the reported results suggest that the proposed solution is not only simple but also effective. Nonetheless, there are aspects of the technical implementation that require additional clarification, and there is room for improvement in the manuscript’s overall clarity and writing style. Given these considerations, my initial assessment leans towards a borderline rating, contingent upon further revision and clarification in the further phases.

**Strengths:**

**[About the idea and motivation]** While the conceptual framework of object-centric reinforcement learning has been explored and analyzed in recent literature, notably by Yoon et al. in 2023, the current manuscript stands out by presenting a harmonious synthesis of object-centric learning and dynamic modeling. The framework demonstrates impressive efficacy, particularly in the realm of pixel-based reinforcement learning control tasks. Consequently, the paper’s core idea holds significant merit and is poised to make a positive impact within the fields of reinforcement learning and dynamic modeling.

**[About the evaluation]** While the existing evaluations substantiate the effectiveness of the proposed methods, there is a need for additional assessments to verify these findings further, details of which I will give in the subsequent sections. Nonetheless, the design and evaluation components of the paper validate the efficacy of the approach, and the conducted ablation studies indicate the significance and impact of each aspect of the model’s design.

**Weaknesses:**

I listed both the weaknesses and questions here.

1. **[About model design]**

- I recognize the potential of employing patch-based object-centric learning for extracting meaningful features for each object. However, there is a concern regarding the sufficiency of these learned factors in accurately modeling the dynamics and policy. Alternatives like deep latent particles [1] could potentially offer more straightforward features, facilitating more effective dynamics modeling. While it is not necessary to conduct experiments with these alternatives during the rebuttal phase, any additional clarification from the authors on how they ensure that the patch-based model can learn representations that are sufficient and robust enough for modeling the dynamics would be highly appreciated and beneficial.

- In Fig. 2 and Section 4.2, it appears that the authors have adopted an assumption of dense interactions among objects, modeled using complete graphs. However, it is a well-acknowledged fact that interactions in real-world scenarios tend to occur sparsely rather than densely. While I know that Graph Neural Networks (GNNs) have the capability to emulate these naturally sparse interactions through their neural representations, there remains a question to me is why the authors chose not to explicitly model these interactions using a sparse graph, which could potentially provide a more direct and accurate representation of the interactions. One reference is [2].

2. Does the proposed method extend its applicability to more realistic scenarios, particularly in terms of generalization capabilities? Specifically, it would be insightful to understand if the method can generalize to scenarios involving a greater number of objects, novel objects, or additional object attributes (such as color) during the inference stage. One reference is [3].

3. **[About presentation]**

The paper demonstrates a clear presentation and logical flow throughout. However, there are several areas that could be enhanced for an even better presentation. Here are some suggestions for improvement:

- I recommend making sections 3.1 and 3.2 to be shorter, given that they cover fundamental concepts in RL that the target audience is likely to be familiar with. By doing so, you can maintain the reader's focus on the core contributions of your work. You might consider moving the more detailed explanations to an appendix.

- For all equations, I suggest employing a different text operation for "node" and "edge", maybe \text{} or \texttt{}, to clearly distinguish these terms from other variables and operations.


- Check the citation formats, use \citep and \citet correctly.

4. The current title, "Graphical Actor Critic,  captures the aspect of the methodology, I recommend incorporating terms like  "Relational" to provide a clearer and more precise indication of the paper's focus. Given that the method does not directly learn or estimate graphical models, this adjustment would help set accurate expectations for the reader and align the title more closely with the paper’s core contributions.

*References*

[1] Daniel, Tal, and Aviv Tamar. "Unsupervised image representation learning with deep latent particles." ICML 2022.

[2] Zadaianchuk, Andrii, Georg Martius, and Fanny Yang. "Self-supervised reinforcement learning with independently controllable subgoals." CoRL 2022.

[3] Zhou, Allan, et al. "Policy architectures for compositional generalization in control." arXiv preprint arXiv:2203.05960 (2022).

**Questions:**

I listed the questions together with weaknesses in the above section.

---

> ### Author Response · Authors · 2023-11-16
> **Response to Reviewer HvnL**
>
> Thank you for your extensive review and for your detailed feedback, this is greatly appreciated.
> Please find our responses to the highlighted weaknesses below.
>
> *[Sufficiency of object-centric representations for RL]* \
> We focus on slot-based object-centric learning models as they aim to extract and represent distinct objects from data into separate slots. Although there is still a gap when it comes to real-world data, STEVE [1] extends SLATE from static images to videos and has been proven to effectively model the temporal dynamics of each slot with good quality. Since the visual complexity of our environments is comparable to the complexity of the environments used to evaluate STEVE, we believe that in our experiments, SLATE representations contain enough information to model the dynamics. \
> [1] Gautam Singh el al. Simple Unsupervised Object-Centric Learning for Complex and Naturalistic Videos. 2022
>
> *[Deep latent particles]* \
> "Deep latent particles" is an interesting approach that is likely more robust for realistic images than SLATE. We trained the model on two datasets containing 100K images from Navigation5x5 and Navigation10x10 tasks, respectively. It accurately detects objects in Navigation5x5 images, but its quality significantly decreases in Navigation10x10 images. We plan to continue experimenting with this model and intend to integrate our approach with this non-slot object-centric model. However, this integration will require further investigation and implementation. We need to find a way to match the object representation from two consecutive states in order to train the transition model.
>
> *[Dense and sparse graphs]* \
> We acknowledge that representing the scene as a complete graph can often result in redundant connections. In the early stages of our research, we explored papers on approaches with sparse interactions between slots and attempted to implement our model with graph sparsification. However, this approach did not yield any improvements and introduced stability issues during training. We appreciate the paper you have suggested, and we will try to incorporate this approach into our model and evaluate its impact. \
> [2] Anirudh Goyal et al. Neural Production Systems: Learning Rule-Governed Visual Dynamics. 2022
>
> *[Generalization capabilities]* \
> The CausalWorld environment, in which the Object Reaching task is implemented, enables the evaluation of an agent's generalization capability across various aspects of the environment. Although we did not specifically focus on the generalization task in this stage of the research, we have plans to conduct it in future stages.
>
> *[Titile]* \
> Is our understanding correct that you are proposing to change the title to "Relational Actor-Critic". We are still thinking about keeping "object-centric" in the title, as it is an important aspect of the model.
>
> *[Comments on writing]* \
> We will incorporate your suggestions into the revised version of the paper, which we will provide before the end of the discussion phase.

---

> > ### Comment · Reviewer_HvnL · 2023-11-20
> >
> > Thanks for the detailed feedback. Some of my concerns have been addressed. It would still be nice to include other object-centric representation learning and sparse graphs in the updated version of this work. I would keep my borderline score for now given the comments/discussions by other reviewers.

---

> > > ### Author Response · Authors · 2023-11-20
> > > **Test of out-of-distribution generalization to unseen colors**
> > >
> > > We evaluated the generalization of the GOCA to unseen colors of distractor objects in the Object Reaching task. When we tested the model with the same colors it was trained on, it achieved a success rate of $0.975 \pm 0.005$. However, when we used new colors for the distractor objects, the success rate dropped to $0.850 \pm 0.005$. The results were averaged over three instances of the GOCA models, and each model was evaluated on 100 episodes. We will further investigate the case when the color of the target object changes.

---

> ### Author Response · Authors · 2023-11-23
>
> We would like to thank you for your helpful critical feedback.
> We have updated our manuscript based on your feedback:
> 1) We changed the title to Relational Object-Centric Actor-Critic.
> 2) We have revised and redesigned the structure of the article based on your feedback (sections 3.1 and 3.2).
> 3) We addressed the questions that were raised during the rebuttal phase.
> 4) The updated parts are highlighted.

---

### Official Review · Reviewer_6gyQ · 2023-11-06

**Soundness:** 2 fair
**Presentation:** 2 fair
**Contribution:** 2 fair
**Rating:** 5
**Confidence:** 4

**Summary:**

The authors presented an object-centric off-policy value-based model-based reinforcement learning approach that uses a pre-trained SLATE model as an object-centric feature extractor; and outperforms object-centric model-free and model-based baselines. Their work shows GNN works well for this type of tasks. If properly presented, the framework presented can become a general purpose method for object-centric downstream tasks in terms of representation.  However, the paper is hard to read and follow and the contributions are not evident.

**Strengths:**

The paper lists down limitations and future scope of work.
The philosophy of using GNN along with Actor Critic is known in other setups, however, the authors have worked hard in the formulations.
It's good to see the details of setup in Appendix.

**Weaknesses:**

The abstract needs to be to the point, crisp, mentioning the the gaps in SOTA and what was done with numbers in support.
The introduction section should contain an illustration of the problem statement and use case.
I believe that this model needs to be tested in embodied ai challenge tasks and their benchmarks to derive the benefits of this in downstream tasks. The work is mostly based on principles of SLATE.
Tuning plays a significant role in GOCA (Fig 6), so the practicality of efficient deployment in a use case is doubtful.
I find the results not substantial.
Supplementary material is not well utilized, no reference to code or videos.

**Questions:**

How does the message passing happen and how does it scale with large graphs?
How are the embedding dimensions come into?
Should the losses be equally weighed? dVAE and cross entropy - after eqn 1.
How do you plan to overcome the listed limitations?

---

> ### Author Response · Authors · 2023-11-16
> **Response to Reviewer 6gyQ**
>
> Thank you for your valuable feedback. Please find our responses to your questions below.
>
> *[How does the message passing happen and how does it scale with large graphs?]* \
> SLATE model produces a set of slots, which are abstract state variables representing objects in the input image. These slots are then processed by GNN-based modules in the GOCA, including the actor model, state-value model, reward model, and transition model. Each GNN consists of two MLPs: an edge model and a node model. The edge model takes a pair of slots as input and produces an edge representation for them. The node model takes a slot as input, along with the sum of representations of all inbound edges, and produces a vector. This process can be seen as a single-step message passing framework, where information is conveyed from neighboring vertices to the current vertex. Since we treat the set of slots as a complete graph, the complexity of the procedure is quadratic in the number of slots, similar to the CSWM [2] transition model. In our experiments, it did not cause performance issues. \
> [1] Gilmer et al. Neural Message Passing for Quantum Chemistry. 2017 \
> [2] Kipf el al. Contrastive Learning of Structured World Models. 2020
>
> *[How are the embedding dimensions come into?]* \
> The embedding dimension is the hyperparameter of the SLATE model, which is called 'Slot Dim' in the SLATE paper [3]. We tuned it to obtain a better representation for Object Reaching and Navigation10x10 environments. However, for Navigation5x5 and PushingNoAgent5x5 environments, we use the same model as they share the observation space. \
> [3] Singh el al. Illiterate DALL-E Learns to Compose. 2022
>
> *[Should the losses be equally weighed? dVAE and cross entropy - after eqn 1.]* (We suppose that you meant equation 11.) \
> We did not use weights for losses but instead assigned different learning rates to the components of the SLATE model (dVAE, encoder, decoder). This is a more flexible approach to balance training between submodules that helps to speed up convergence.
>
> *[How do you plan to overcome the listed limitations?]*
> 1. The first limitation is a deterministic world model, which may struggle to predict the dynamics of highly stochastic environments. To address the issue, we plan to develop an object-centric world model algorithm based on Dreamer V3, where the model explicitly maintains the deterministic and stochastic parts of the state.
> 2. Entropy tuning. Our approach is based on the SAC algorithm. It is a known issue that discrete SAC is sensitive to entropy-related hyperparameters. We plan to adopt a metagradient-based approach [4] that does not require the target entropy hyperparameter. \
> [4] Yufei Wang, Tianwei Ni. Meta-SAC: Auto-tune the Entropy Temperature of Soft Actor-Critic via Metagradient. 2020

---

> > ### Author Response · Authors · 2023-11-16
> >
> > Our responses on the highlighted weaknesses of the work.
> >
> > *[I believe that this model needs to be tested in embodied ai challenge tasks and their benchmarks to derive the benefits of this in downstream tasks.]* \
> > Embodied AI environments like Habitat [5] have visually rich realistic oberservation space which presents difficult challange for the current unsupervised slot-based object-centric models. On such datasets, object-centric models often get stuck in a failure mode where they cannot bind slots to objects in the input. We plan to integrate the recent DINOSAUR [6] model into GOCA, which demonstrates promising results on realistic datasets. However, in this work, our main focus is on exploring how slot-based object-oriented representations can be used in model-based RL. \
> > [5] Szot el al. Habitat 2.0: Training Home Assistants to Rearrange their Habitat. 2021 \
> > [6] Seitzer el al. Bridging the Gap to Real-World Object-Centric Learning. 2023
> >
> > *[The work is mostly based on principles of SLATE.]* \
> > While we use the SLATE model as an object-centric feature extractor, in principle, we can replace SLATE with other slot-based object-centric models, since the GNN-based modules expect a set of slots as input.
> >
> > *[I find the results not substantial.]* \
> > We consider the primary contribution of our work to be the proposal of a novel actor-critic architecture that uses an object-centric world model as a component of the critic. This architecture aims to improve the quality of the predictions made by the critic and demonstrate its positive impact on the sample efficiency of policy learning. We extended the SAC algorithm by integrating the world model into the critic and introducing a new objective function to train it. Additionally, we modified the CSWM [7] transition model by adjusting its edge model, where we pass a pair of slots into the edge model along with an action. Our results indicate that in environments where current object-centric representation models perform well, the proposed algorithm performs on par with model-free algorithms in simple tasks like Navigation5x5. However, it scales better as the number of objects increases (Navigation10x10) or the environment dynamics become more challenging (PushingNoAgent5x5). We believe that extending the expressiveness of object-centric representation models or addressing the hyperparameter sensitivity of discrete SAC are important lines of work, but we focus on object-centric model-based RL research. \
> > [7] Kipf el al. Contrastive Learning of Structured World Models. 2020
> >
> > *[Comments on writing and supplementary materials]* \
> > We will consider your feedback regarding the writing style and supplementary materials and will revise the paper before the end of the discussion phase.

---

> > > ### Comment · Reviewer_6gyQ · 2023-11-21
> > > **Response to rebuttal**
> > >
> > > Thanks to the authors for the responses. Make sure that novelty aspect comes out clearly. Waiting for the unaddressed points to update my score accordingly later.

---

> ### Author Response · Authors · 2023-11-23
>
> We would like to thank you for your helpful critical feedback.
> We have updated our manuscript based on your feedback:
> 1) We added the illustration for our method to the introduction.
> 2) We added the source code and videos to the supplementary materials.
> 3) We addressed the questions that were raised during the rebuttal phase.
> 4) The updated parts are highlighted.

---

### Meta-Review · Area_Chair_s7N6 · 2023-12-06

**Metareview:**

The authors propose a reinforcement learning algorithm that learns a graph neural network transition function from pixels to then predict Q values for learning a policy. The paper aims to address a promising research questions, shows some promise in the experiments, and improved during the discussion period, but the empirical performance is only marginally better than baselines. Some claims are overstated, e.g. the caption of Fig 4 mentions faster learning than baselines which is not supported by the curves in the figure. Choosing tasks that clearly demonstrate the benefit of the object-centric representations and generally improved empirical performance would strengthen the paper.

**Justification For Why Not Higher Score:**

The empirical gains are marginal

**Justification For Why Not Lower Score:**

N/A

---

### Decision · Program_Chairs · 2024-01-16

Reject